# The efficacy and safety of acetazolamide in chronic mountain sickness: A systematic review and meta-analysis of randomized controlled trials

**Yaqin Wang[1], Zhengcai Han[1], Zhouzhou Feng[1,2]***

**1** The First Hospital of Lanzhou University, Lanzhou City, Gansu Province, People's Republic of China, **2** Lanzhou University, The First Clinical Medical College of Lanzhou University, Lanzhou City, Gansu Province, People's Republic of China

* fengzhzh21@126.com

## Abstract

### Objective

The impact of acetazolamide (ACZ) in chronic mountain sickness (CMS) has not been fully assessed. The purpose of this systematic review is to evaluate the effectiveness and safety of acetazolamide in the treatment of chronic mountain sickness.

### Research methods

This systematic review and meta-analysis were conducted following the Preferred Reporting Items for Systematic Reviews and Meta-Analyses (PRISMA) guidelines. The primary outcome measure was CMS clinical score. Secondary outcomes included CMS total score, hematocrit (HCT), Pondus Hydrogenii (pH), arterial oxygen pressure (PaO2), arterial carbon dioxide pressure (PaCO2), bicarbonate concentration (HCO3), and adverse events.

### Results

Five randomized controlled trials were included, comprising a total of 137 subjects, with 78 in the acetazolamide group and 59 in the control group. The CMS clinical score showed an MD of −0.31 (95% CI, −1.13 to −0.51, P = 0.46), the results indicated no statistical significance. But the CMS total score had an MD of −1.13 [95% CI, −2.03 to −0.23], P = 0.01, showing a significant difference. The HCT results showed an MD of −2.70 (95% CI, −4.58 to −0.82; P = 0.005), indicating a statistically significant reduction. The result of PaO2, PaCO2, pH and HCO3 are statistically significant. In terms of adverse events, increased diuresis and headache were not statistically significant. Paresthesia had a significant difference.

### Conclusion

Based on the available evidence, we conclude that ACZ 250 mg is a safe, reliable, and low-cost treatment option for chronic mountain sickness. By reducing HCT, PaCO2, pH,

**Data availability statement:** All relevant data are within the paper and its Supporting Information files.

**Funding:** The present study was supported by the Project for the First Hospital Fund of Lanzhou University (grant no. ldyyyn2023-56). The funders played a leading role in this study.

**Competing interests:** The authors have declared that no competing interests exist

and $HCO_3$, and increasing $PaO_2$, it improves respiratory and circulatory parameters in CMS patients and effectively treats CMS.

## Introduction

Chronic mountain sickness (CMS), also known as Monge's disease, was first described in 1928 by Carlos Monge from Peru [1]. In 1998, the International CMS Consensus Group defined it as a syndrome characterized by prolonged exposure to high altitudes, where low oxygen levels lead to compensatory overproduction of red blood cells, also known as excessive erythrocytosis(EE) (defined as hemoglobin concentration [Hb] ≥ 21 g/dL in men and ≥ 19 g/dL in women) and an increase in hematocrit (level ≥ 63% for males and ≥ 57% for females), resulting in significantly increased blood viscosity, microcirculation disturbances, and even widespread organ damage and reduced blood flow velocity [2,3]. Additionally, the low-pressure, hypoxic environment at high altitudes reduces arterial oxygen pressure and arterial oxygen saturation (SaO2), leading to decreased tissue oxygenation, cellular hypoxia, and potential organ dysfunction [4,5]. When hypoxia-induced factor in the kidneys are stimulated by hypoxia, interstitial fibroblasts in the renal tubules secrete erythropoietin (EPO) to stimulate hematopoiesis in the bone marrow, promoting the division of nucleated red blood cells and accelerating red blood cell maturation, thereby increasing the number of red blood cells in the blood [6]. This condition leads to diseases characterized by high viscosity syndrome, including high-altitude polycythemia (HAPC), high-altitude pulmonary hypertension (HAPH), and high-altitude heart disease (HAHD), which pose serious health risks to high-altitude residents [7,8].

Approximately 140 million people worldwide live in high-altitude regions (>2500 meters), mainly concentrated in South America (Andes), Central Asia (Tibetan and Sherpa populations), and East Africa (Ethiopia) [9]. Among them, 5% to 10% of people develop chronic mountain sickness (CMS) due to an inability to adapt to chronic hypoxia. The main clinical symptoms of CMS include headache, dizziness, tinnitus, shortness of breath, palpitations, sleep disturbances, and cognitive deficits [10]. The pathogenesis of CMS is complex, with various risk factors that predispose individuals to the disease, possibly related to ethnicity, age, sex, altitude, and genetic susceptibility [11–13]. Despite various pharmacological and non-pharmacological interventions tested for EE and CMS, the optimal treatment method remains to be determined [14,15].

Acetazolamide (ACZ) is a carbonic anhydrase inhibitor that works by inhibiting the hydration of CO2, reducing HCO3- reabsorption, increasing H + and CO2 levels in the blood and tissues (including cerebrospinal fluid), stimulating central chemoreceptors (CCR) to enhance respiration, increasing lung ventilation, oxygen pressure, and oxygen saturation, while decreasing carbon dioxide pressure [16,17]. ACZ has been approved by the U.S. Food and Drug Administration (FDA) for the prevention and treatment of acute mountain sickness (AMS) [18,19]. Numerous studies have shown that ACZ is effective in preventing AMS in healthy lowlanders traveling to high-altitude regions [20,21], and it also prevents adverse health effects in patients with chronic obstructive pulmonary disease (COPD) due to altitude [22,23]. Animal studies have shown that ACZ can effectively treat CMS by improving erythropoiesis, blood viscosity, pulmonary circulation, ventilation, and cardiac function, among other physiological and pathological aspects [24]. Clinical studies have demonstrated similar results, indicating that ACZ (250 mg/day) should be a low-cost treatment option for chronic mountain sickness [25–30].

Despite the promising results of ACZ in CMS studies, there remains a lack of solid evidence-based support for its use in CMS. Therefore, we conducted this systematic review

and meta-analysis, incorporating the latest published literature to evaluate, for the first time, the efficacy and safety of ACZ in treating CMS. Our goal is to provide reliable evidence-based clinical data for assessing the effectiveness of ACZ in the management of CMS.

## Methods

This systematic review and meta-analysis was registered at PROSPERO (http://www.crd.york.ac.uk/prospero, CRD: 42023466072) and designed as per the Cochrane Handbook for Systematic Reviews of Interventions [31]. The reporting followed the PRISMA guidelines [32].

### Data sources and searches

The databases searched for this review included PubMed, Web of Science, Embase, and the Cochrane Library. The search covered all publications from the inception of these databases up to August 30, 2024. The search strategy combined subject headings and free-text terms related to "Acetazolamide" and "Chronic mountain sickness." Detailed search strategies can be found in the S1 Appendix.

### Literature data inclusion and exclusion criteria

**Inclusion criteria:**

1. Adults who reside in high-altitude areas ($\geq$ 2500 meters) for a long time(at least 1 year).

2. Participants who meet the diagnostic criteria for chronic mountain sickness.

3. Studies with a randomized controlled trial (RCT) design.

4. Studies that provide at least one data outcome of interest for extraction.

**Exclusion criteria:**

1. Participants with other acute or chronic illnesses.

2. Animal experiments.

3. Studies with incomplete data or data that cannot be extracted.

4. Non-English literature.

5. Studies where the full text is not available.

### Types of outcome measures

The primary outcome indicator selected was CMS clinical score. Secondary outcomes included CMS total score,hematocrit (HCT), Pondus Hydrogenii (pH), arterial oxygen pressure (PaO2), arterial carbon dioxide pressure (PaCO2), bicarbonate concentration (HCO3), and adverse events.

### Data extraction and quality assessment

We imported the retrieved literature into ENDNOTE X9 software. Two researchers independently reviewed each document according to the inclusion and exclusion criteria. For trials that met the inclusion criteria, we extracted basic information from the articles, such as the first author's last name, year of publication, type of participant, sample size, interventions, controls, and outcomes. The quality of randomized controlled trials was assessed using the Cochrane Risk of Bias tool in Review Manager 5.4 software (Cochrane Collaboration, Oxford, England)

which includes criteria such as random sequence generation, allocation concealment, blinding of participants and personnel, blinding of outcome assessment, incomplete outcome data, selective reporting, and other potential biases. Each item was rated as "low risk," "high risk," or "unclear." Disagreements were resolved through arbitration by a third investigator.

## Statistical analysis

All analyses were performed using Review Manager 5.4. Dichotomous variables were expressed as risk ratios (RR), continuous variables as mean differences (MD), and each effect size was reported with a 95% confidence interval (CI). Heterogeneity between study outcomes was analyzed using the $I^2$ test. We interpreted I2 using the guidance available from the Cochrane handbook [30]. A fixed-effects model was used when the test for heterogeneity had $P \geq 0.05$ and $I^2 < 50\%$. If $P < 0.05$ and $I^2 \geq 50\%$, a random-effects model was applied. A P-value of $< 0.05$ was considered statistically significant. The Engauge Digitizer (version 4.1) graphical data extraction software was used to extract data provided by images only. Funnel plots and Egger tests were not used to assess potential publication bias due to the small number of included studies (<10). Sensitivity analyses were performed on outcome metrics with high heterogeneity to determine the stability of the results.

## Results

Five randomized controlled studies were included in this analysis [26–30], comprising a total of 137 subjects, with 78 in the ACZ group and 59 in the control group. The screening process is illustrated in Fig 1, and the basic characteristics of the included studies are summarized in Table 1.

Among the five included studies, one study [27] did not specify the method of generating the randomized sequence or indicate whether blinding was used. In this study, the control group received nocturnal oxygen treatment. The other four studies [26,28–30] specified the implementation of randomized sequences and blinding, with the control group treated with a placebo. No other significant risks of bias were identified in any of the studies. Detailed risk of bias assessments are presented in Fig 2.

### Primary outcome

**CMS clinical score.** Four studies [26,27,29,30] reported CMS clinical scores. The heterogeneity test showed $I^2 = 1\%$, $P = 0.39$. Using a fixed-effects model, the results indicated an MD of -0.31 [95% CI, -1.13 to -0.51], $P = 0.46$. Based on the 95% confidence intervals, the results showed no statistical significance, indicating that ACZ cannot improve the clinical symptoms of CMS patients (Fig 3).

### Secondary outcomes

**CMS total score.** Three studies [26–28] reported CMS total scores. The heterogeneity test showed $I^2 = 0\%$, $P = 0.74$. Using a fixed-effects model, the results indicated an MD of -1.13 [95% CI, -2.03 to -0.23], $P = 0.01$. Based on the 95% confidence intervals, the difference was considered statistically significant, suggesting that ACZ effectively improves the CMS total score (Fig 4).

### Laboratory parameters

**HCT.** Five studies [26–30] reported HCT results. The heterogeneity test showed $I^2 = 30\%$, $P = 0.22$. Using a fixed-effects model, the results demonstrated a statistically significant difference in HCT with a mean difference (MD) of -2.70 [95% CI, -4.58 to -0.82], $P = 0.005$, suggesting that ACZ significantly reduces HCT in CMS patients (Fig 5).

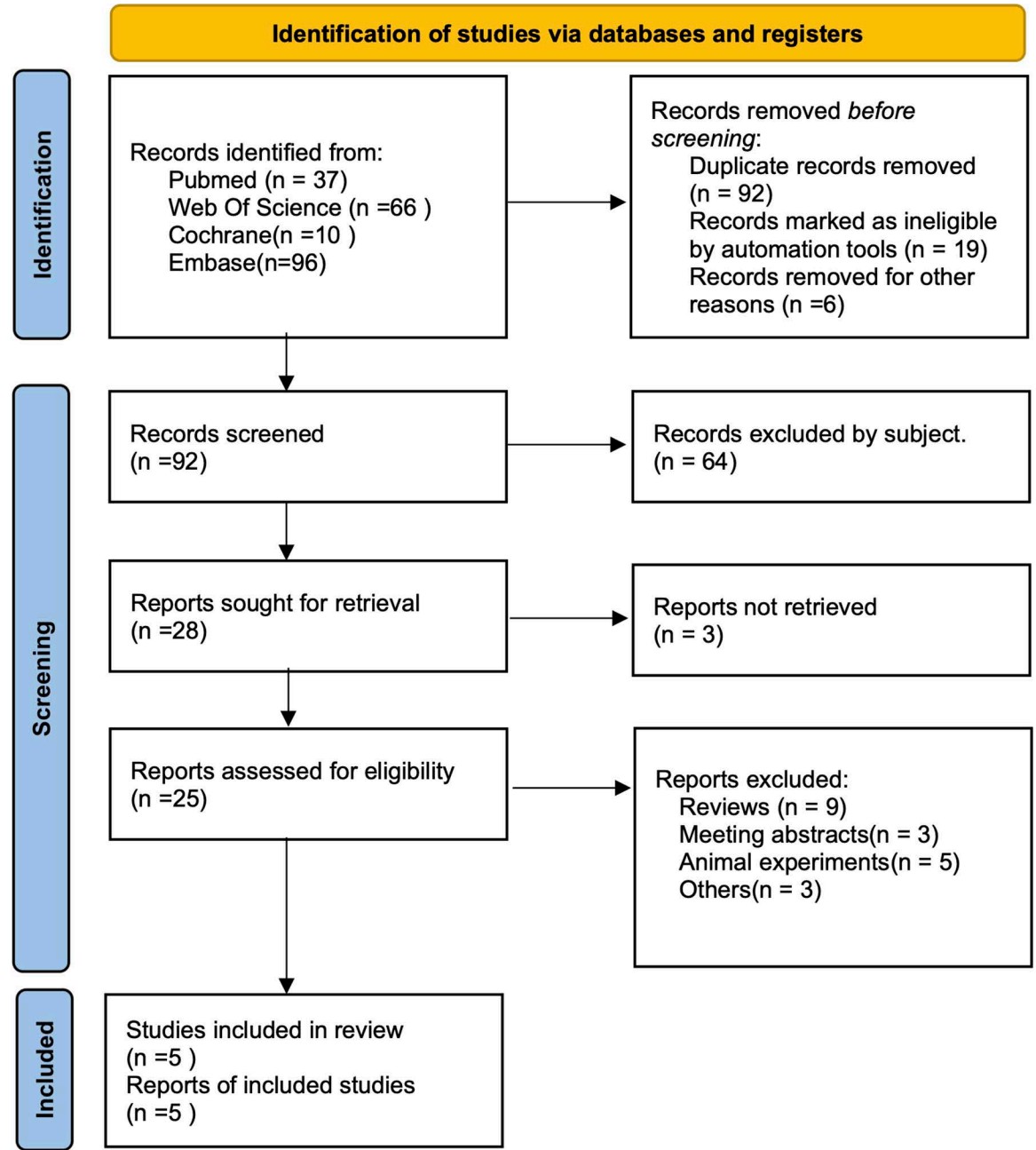

**Fig 1. PRISMA (preferred reporting items for systematic reviews and meta-analysis) flow diagram.**

**PaO2.** Five studies [26–30] reported PaO2 results. The heterogeneity test showed I² = 27%, P = 0.24. Using a fixed-effects model, the results indicated a statistically significant difference with an MD of 2.00 [95% CI, 0.77 to 3.22], P = 0.001, suggesting that ACZ effectively increases arterial oxygen levels in CMS patients ([Fig 6]).

**PaCO2.** Five studies [26–30] reported $PaCO_2$ results. The heterogeneity test showed I² = 27%, P = 0.24. Using a fixed-effects model, the results demonstrated an MD of -3.27 [95% CI, -4.16 to -2.39], P < 0.00001, suggesting that ACZ effectively reduces $CO_2$ retention in CMS patients ([Fig 7]).

**Table 1. Characteristics of all studies included in meta-analysis.**

| Auther Year | Methods | Intervention | | Sample size | | Mean age (years) | | CMS total score | | Altitude | Duration |
|---|---|---|---|---|---|---|---|---|---|---|---|
| | | Treatment | Control | Treatment | Control | Treatment | Control | Treatment | Control | | |
| Champigneulle B 2023 [26] | RCT | acetazolamide 250mg/d | placebo | 13 | 14 | 45 ± 8 | 46 ± 8 | 8 ± 1 | 9 ± 3 | 5100-5300m | 3W |
| Schmidt WFJ 2023 [27] | RCT | acetazolamide 250mg/d | Night oxygen supply | 7 | 5 | 54.0 ± 5.4 | 49.27 ± 0.0 | 9.7 ± 3.8 | 9.2 ± 3.6 | 3900m | 3W |
| Sharma S 2017 [28] | RCT | acetazolamide 250mg/d | placebo | 15 | 17 | 46.6 ± 8.9 | 46.5 ± 11.39 | 10.3 ± 3.1 | 9.5 ± 2.3 | 4380m | 6W |
| Richalet JP 2008 [29] | RCT | acetazolamide 250mg/d | placebo | 34 | 13 | 44 ± 10 | 48 ± 11 | 12 ± 3 | | 4,300m | 12W |
| Richalet JP 2005 [30] | RCT | acetazolamide 250mg/d | placebo | 9 | 10 | 43 ± 9 | 44 ± 9 | 22.8 ± 9.8 | | 4,300m | 3W |

Abbreviations: RCT: randomized controlled trial mg: milligram d:day m:metre W:week.

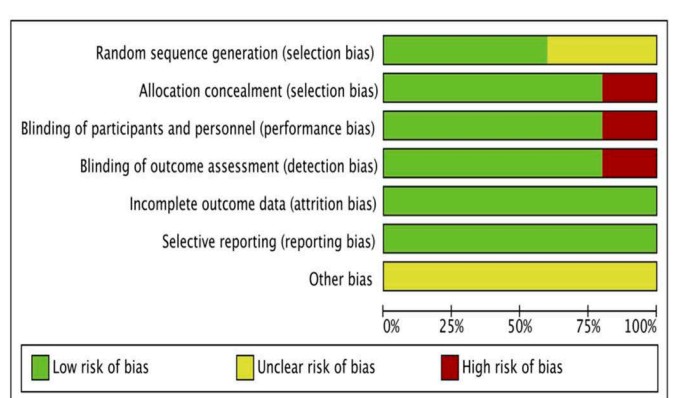

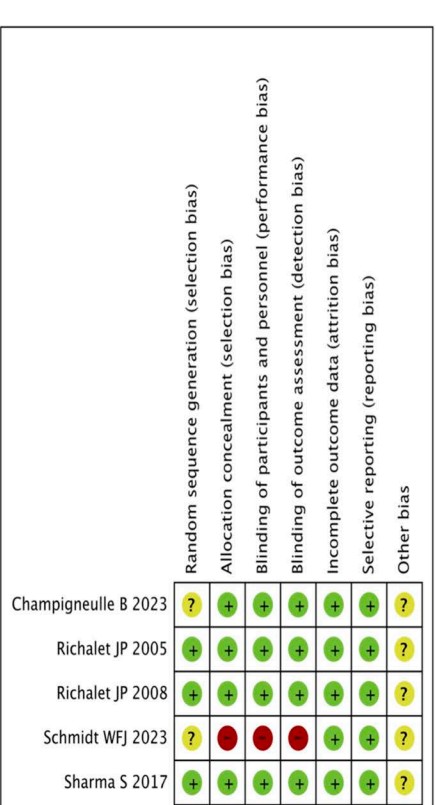

**Fig 2. Risk of bias across studies.**

**pH.** All five studies [26–30] reported pH results. The heterogeneity test showed $I^2 = 86\%$, $P < 0.00001$. Using a random-effects model, the results showed an MD of -0.07 [95% CI, -0.11 to -0.03], $P = 0.0002$, indicating a statistically significant reduction in pH. Due to high heterogeneity, a sensitivity analysis was performed, which suggested that removing the study by Richalet et al. (2005) decreased heterogeneity to $I^2 = 0\%$, MD -0.09 [95% CI, -0.11 to -0.07], $P < 0.00001$, and the results remained unchanged (Fig 8).

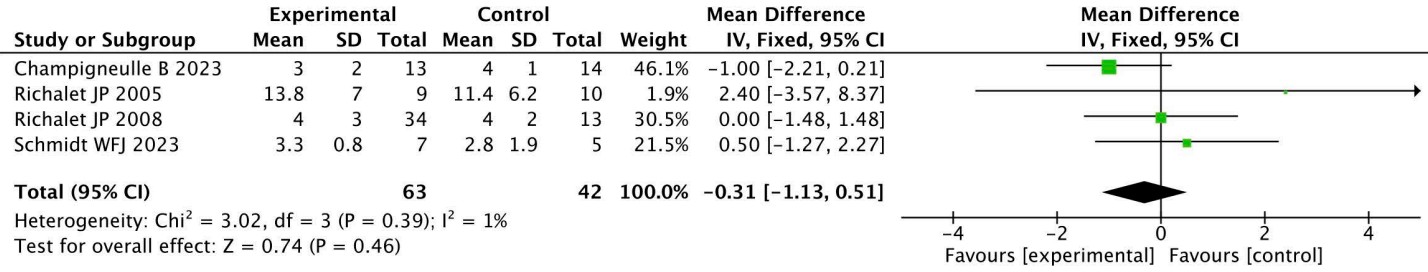

**Fig 3. Forest plot of CMS clinical score.**

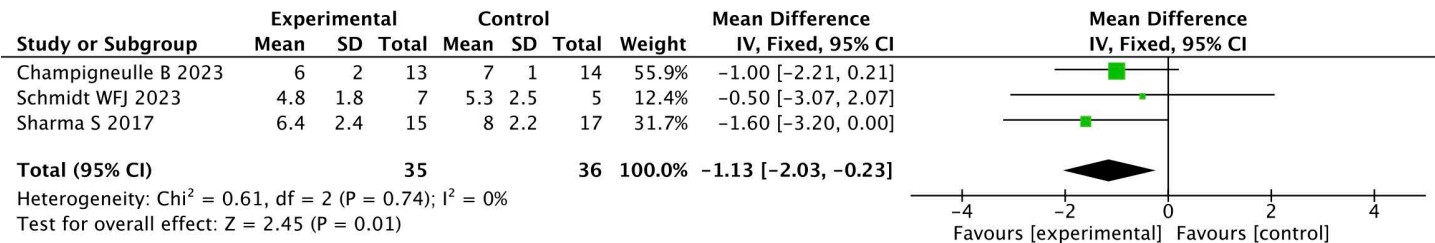

**Fig 4. Forest plot of CMS total score.**

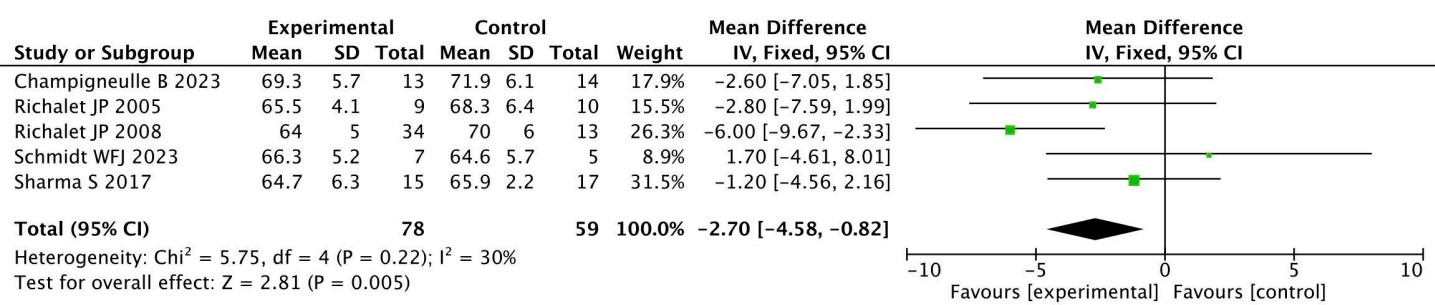

**Fig 5. Forest plot of HCT.**

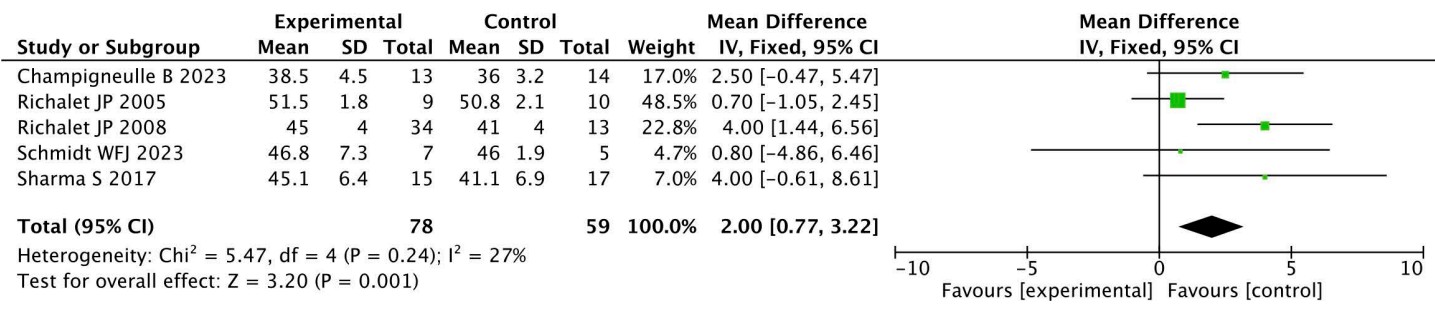

**Fig 6. Forest plot of PaO2.**

**HCO3.** Four studies [27–30] reported HCO3 results. The heterogeneity test showed $I^2 = 79\%$, P = 0.003. Using a random-effects model, the results indicated an MD of -4.59 [95% CI, -6.35 to -2.83], P < 0.00001, suggesting that ACZ effectively reduces $HCO_3$ in CMS patients. Sensitivity analysis indicated that the results were stable and unchanged when any of the studies were removed (Fig 9).

**Adverse events.** Four studies [26,28–30] reported adverse events. One study (Richalet et al., 2005) specified that there was no statistical difference in adverse events between the experimental and control groups but did not provide specific data.

**Increased diuresis.** Two studies [29,30] reported increased diuresis in CMS patients. The heterogeneity test showed $I^2 = 0\%$, P = 0.059. Using a fixed-effects model, the results showed an RR of 1.52 [95% CI, 0.87 to 2.66], P = 0.14, indicating no statistically significant difference (Fig 10A).

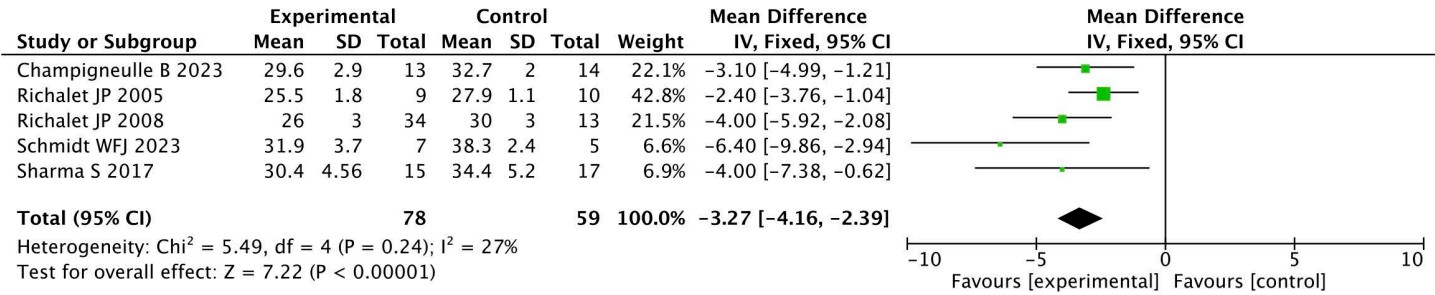

**Fig 7. Forest plot of PaCO2.**

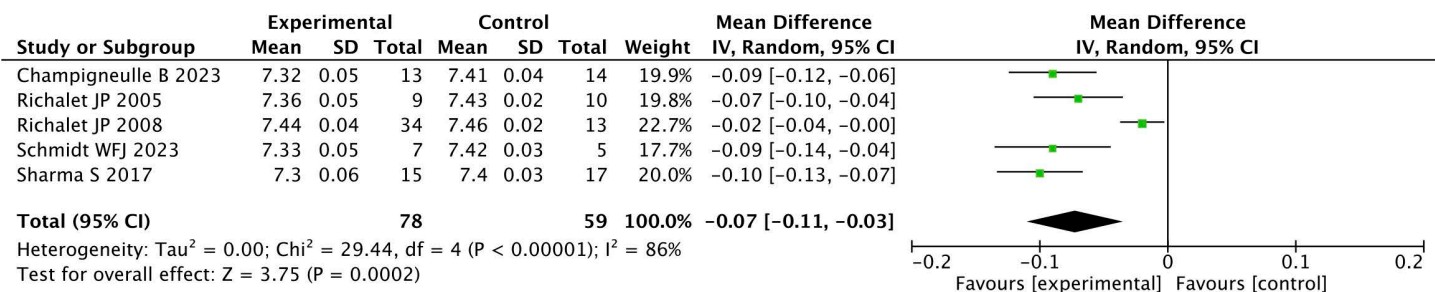

**Fig 8. Forest plot of pH.**

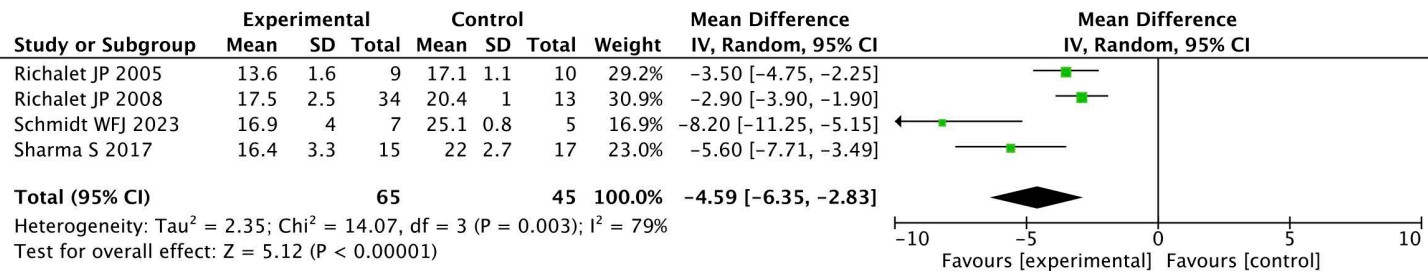

**Fig 9. Forest plot of HCO3.**

**Paresthesia.** Three studies [28–30] reported paresthesia in CMS patients. The heterogeneity test showed $I^2 = 0\%$, P = 0.93. Using a fixed-effects model, the results showed an RR of 1.82 [95% CI, 1.02 to 3.25], P = 0.04, indicating a statistically significant difference (Fig 10B).

**Headache.** Two studies [28,29] reported headaches in CMS patients. The heterogeneity test showed $I^2 = 0\%$, P = 0.85. Using a fixed-effects model, the results showed an RR of 0.49 [95% CI, 0.21 to 1.13], P = 0.09, indicating no statistically significant difference (Fig 10C).

## Discussion

This study includes randomized controlled literature on the efficacy of ACZ 250mg in treating CMS. A systematic and comprehensive analysis of the extracted data was conducted through meta-analysis, providing strong evidence-based support for the clinical treatment of CMS. Our results indicate that ACZ significantly reduced HCT, PaCO2, pH, and HCO3, while increasing PaO2. These effects may be attributed to the pharmacological actions of ACZ, which include increasing central chemoreceptor sensitivity, inhibiting the carotid body, and stimulating the respiratory center, leading to overall increased ventilation, reduced renal EPO production, and decreased erythropoiesis and hemoglobin levels [33]. The reduction in HCT may also be related to ACZ's ability to increase plasma volume (PV) through fluid regulation [34]. Chronic hypoxia reduces PV, while increased red blood cell volume increases total blood volume [35,36]. Experiments show that compared to hypoxic control rats, hypoxic rats treated with ACZ had a slight increase in PV, which may reflect physiological adaptations to maintain

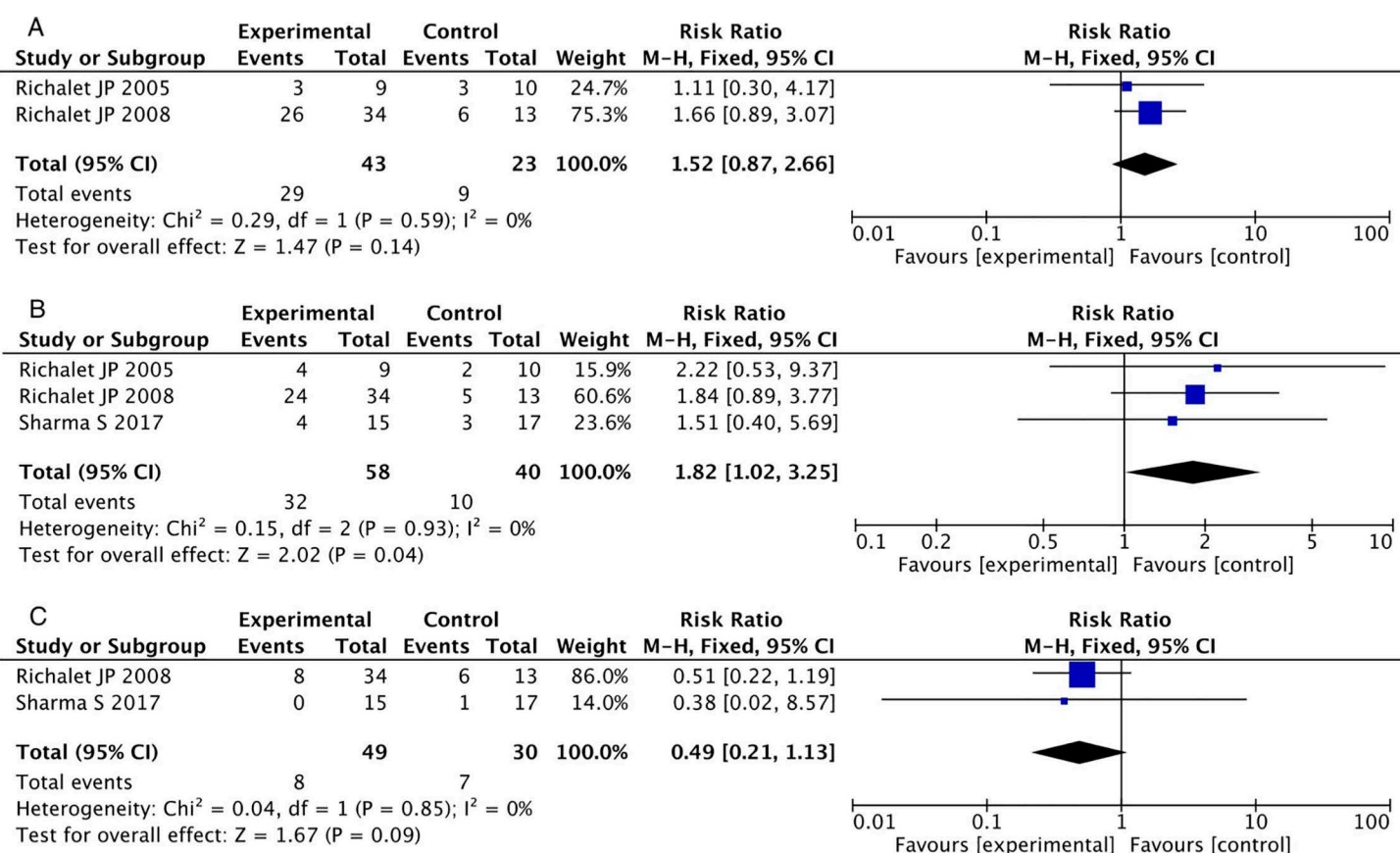

**Fig 10. Forest plot of adverse event.** (A) Increased diuresis (B) paresthesia (C) Headache.

normal blood volume and the effects of ACZ [24]. Whole blood viscosity and its oxygen-binding capacity are primarily determined by HCT, which increases exponentially with higher HCT and oxygen capacity [37]. ACZ treatment effectively reduced HCT and blood viscosity, and ACZ increased ventilation and improved alveolar oxygen pressure, increasing PaO2 and reducing PaCO2, thereby decreasing hypoxic pulmonary vasoconstriction and limiting pulmonary artery remodeling and vascular resistance [38]. ACZ also improved ventilation/perfusion matching, enhancing the efficiency of pulmonary gas exchange and hemodynamic oxygen delivery, reducing hypoxic stress [39].

In the analysis of CMS scores, we found no statistically significant difference in clinical CMS scores between the experimental and control groups. However, interestingly, there was a significant difference in the overall CMS score, suggesting that ACZ's effect on improving clinical symptoms in CMS patients was not as expected. It is well known that the overall CMS score is calculated based on hemoglobin concentration and seven persistent clinical symptoms (fatigue and palpitations, insomnia, cyanosis, vasodilation, paresthesia, headache, and tinnitus) [10]. The clinical CMS score, however, does not account for hemoglobin concentration. Therefore, ACZ may improve patients' overall CMS score, possibly because it improves oxygenation of tissues and organs, thereby reducing hemoglobin levels. Notably, ACZ's side effects include increased leg fatigue during hypoxic exercise, impaired respiratory muscle function, and paresthesia [40,41]. These side effects overlap with CMS symptoms, complicating the determination of their cause. The CMS score may be too simplistic to clearly demonstrate treatment efficacy, functioning more as an epidemiological tool to assess CMS prevalence in populations. Additionally, subjective symptom scoring systems are susceptible to placebo effects, and a new clinical CMS scoring system should be developed for interventional studies [29].

In the evaluation of adverse events, the most commonly reported reactions were polyuria, paresthesia, and headache. Given ACZ's diuretic properties, polyuria was expected, but our results showed no statistical difference between the experimental and control groups. This may be because CMS patients often experience polyuria to reduce blood volume and cardiac load. Fortunately, no electrolyte disturbances or organ damage due to polyuria occurred during treatment. Paresthesia was significantly higher in the ACZ group, which could be attributed to ACZ itself or a worsening of CMS symptoms. There was no significant difference in headache incidence, suggesting that ACZ has limited efficacy in reducing CMS-related headaches. Additionally, research has shown that the prevalence of CMS varies significantly between genders, possibly due to the regulatory effects of estrogen on erythropoiesis [42]. Overall, ACZ did not cause any serious adverse events, indicating its relative safety. However, the current treatment duration was relatively short, and gender differences were not accounted for. Given the chronic nature of CMS treatment, further research is needed to clarify the long-term efficacy and side effects of ACZ treatment in different populations, at different altitudes, and with varying doses.

## Limitations

Although the literature included in this study is generally of high quality and the results are relatively reliable, there are still certain limitations. First, the number of studies included and the sample size were small. Second, graphic data extraction software was used to extract data provided only in images; while this method is commonly used in meta-analyses, it may introduce some errors. Third, this study only analyzed the efficacy of ACZ at a dose of 250 mg/day. Due to limited data, lower doses (125 mg/day) and higher doses (500 mg/day) were not evaluated. Fourth, this study only investigated the short-term efficacy and adverse effects of ACZ in treating CMS, and the long-term efficacy and adverse effects remain unclear due to

limited data. Therefore, more high-quality, large-sample, multi-center randomized studies are needed in the future to further validate the efficacy and safety of different doses of ACZ and long-term treatment for CMS.

## Conclusion

Based on the available evidence, we conclude that ACZ 250 mg is a safe, reliable, and low-cost treatment option for chronic mountain sickness. By reducing HCT, PaCO2, pH, and HCO3, and increasing PaO2, it improves respiratory and circulatory parameters in CMS patients and effectively treats CMS.

## Supporting information

**S1 Appendix. Search strategy.**
(DOCX)

**S1 File. Search result.**
(XLSX)

**S1 Data. All data extracted.**
(XLSX)

**S1 Checklist. PRISMA checklist.**
(DOCX)

**S1 Table. GRADE analysis.**
(DOCX)

## Acknowledgments

The authors gratefully acknowledge all the authors who shared the results of the included studies, the support of the Research Center for Clinical Medicine, The First Hospital of Lanzhou University.

## Author contributions

**Conceptualization:** Yaqin Wang, Zhengcai Han, Zhouzhou Feng.

**Data curation:** Yaqin Wang, Zhengcai Han, Zhouzhou Feng.

**Formal analysis:** Yaqin Wang, Zhouzhou Feng.

**Funding acquisition:** Yaqin Wang, Zhouzhou Feng.

**Investigation:** Yaqin Wang, Zhouzhou Feng.

**Methodology:** Yaqin Wang, Zhengcai Han, Zhouzhou Feng.

**Project administration:** Yaqin Wang, Zhouzhou Feng.

**Resources:** Yaqin Wang, Zhouzhou Feng.

**Software:** Yaqin Wang, Zhengcai Han, Zhouzhou Feng.

**Supervision:** Yaqin Wang, Zhengcai Han, Zhouzhou Feng.

**Validation:** Yaqin Wang, Zhouzhou Feng.

**Visualization:** Yaqin Wang, Zhouzhou Feng.

**Writing – original draft:** Yaqin Wang, Zhouzhou Feng.

**Writing – review & editing:** Yaqin Wang, Zhouzhou Feng.

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
