## [Decision Letter · Decision Letter 0]

11 Dec 2024

PONE-D-24-42748The efficacy and safety of acetazolamide in chronic mountain sickness：a systematic review and meta‐analysis of randomized controlled trialsPLOS ONE

Dear Dr. Feng,

Thank you for submitting your manuscript to PLOS ONE. After careful consideration, we feel that it has merit but does not fully meet PLOS ONE’s publication criteria as it currently stands. Therefore, we invite you to submit a revised version of the manuscript that addresses the points raised during the review process.

We look forward to receiving your revised manuscript.

Kind regards,

Tungki Pratama Umar, M.D.

Academic Editor

PLOS ONE

Journal Requirements: When submitting your revision, we need you to address these additional requirements. 1. Please ensure that your manuscript meets PLOS ONE's style requirements, including those for file naming. The PLOS ONE style templates can be found at https://journals.plos.org/plosone/s/file?id=wjVg/PLOSOne_formatting_sample_main_body.pdf and https://journals.plos.org/plosone/s/file?id=ba62/PLOSOne_formatting_sample_title_authors_affiliations.pdf 2. As required by our policy on Data Availability, please ensure your manuscript or supplementary information includes the following:  A numbered table of all studies identified in the literature search, including those that were excluded from the analyses.   For every excluded study, the table should list the reason(s) for exclusion.   If any of the included studies are unpublished, include a link (URL) to the primary source or detailed information about how the content can be accessed.  A table of all data extracted from the primary research sources for the systematic review and/or meta-analysis. The table must include the following information for each study:  Name of data extractors and date of data extraction  Confirmation that the study was eligible to be included in the review.   All data extracted from each study for the reported systematic review and/or meta-analysis that would be needed to replicate your analyses.  If data or supporting information were obtained from another source (e.g. correspondence with the author of the original research article), please provide the source of data and dates on which the data/information were obtained by your research group.  If applicable for your analysis, a table showing the completed risk of bias and quality/certainty assessments for each study or outcome.  Please ensure this is provided for each domain or parameter assessed. For example, if you used the Cochrane risk-of-bias tool for randomized trials, provide answers to each of the signalling questions for each study. If you used GRADE to assess certainty of evidence, provide judgements about each of the quality of evidence factor. This should be provided for each outcome.   An explanation of how missing data were handled.  This information can be included in the main text, supplementary information, or relevant data repository. Please note that providing these underlying data is a requirement for publication in this journal, and if these data are not provided your manuscript might be rejected.  3. Thank you for stating the following financial disclosure: "The present study was supported by the Project for the First Hospital Fund of Lanzhou University (grant no. ldyyyn2023-56)." Please state what role the funders took in the study.  If the funders had no role, please state: ""The funders had no role in study design, data collection and analysis, decision to publish, or preparation of the manuscript."" If this statement is not correct you must amend it as needed. Please include this amended Role of Funder statement in your cover letter; we will change the online submission form on your behalf. 4. Thank you for stating the following in the Acknowledgments Section of your manuscript: "The authors gratefully acknowledge all the authors who shared the results of the included studies, the support of the Research Center for Clinical Medicine,The First Hospital of Lanzhou University.The present study was supported by the Project for the First Hospital Fund of Lanzhou University (grant no. ldyyyn2023-56)." We note that you have provided funding information that is not currently declared in your Funding Statement. However, funding information should not appear in the Acknowledgments section or other areas of your manuscript. We will only publish funding information present in the Funding Statement section of the online submission form. Please remove any funding-related text from the manuscript and let us know how you would like to update your Funding Statement. Currently, your Funding Statement reads as follows: "The present study was supported by the Project for the First Hospital Fund of Lanzhou University (grant no. ldyyyn2023-56)." Please include your amended statements within your cover letter; we will change the online submission form on your behalf. 5. We note that your Data Availability Statement is currently as follows: All relevant data are within the manuscript and its Supporting Information files. Please confirm at this time whether or not your submission contains all raw data required to replicate the results of your study. Authors must share the “minimal data set” for their submission. PLOS defines the minimal data set to consist of the data required to replicate all study findings reported in the article, as well as related metadata and methods (https://journals.plos.org/plosone/s/data-availability#loc-minimal-data-set-definition). For example, authors should submit the following data: - The values behind the means, standard deviations and other measures reported;- The values used to build graphs;- The points extracted from images for analysis. Authors do not need to submit their entire data set if only a portion of the data was used in the reported study. If your submission does not contain these data, please either upload them as Supporting Information files or deposit them to a stable, public repository and provide us with the relevant URLs, DOIs, or accession numbers. For a list of recommended repositories, please see https://journals.plos.org/plosone/s/recommended-repositories. If there are ethical or legal restrictions on sharing a de-identified data set, please explain them in detail (e.g., data contain potentially sensitive information, data are owned by a third-party organization, etc.) and who has imposed them (e.g., an ethics committee). Please also provide contact information for a data access committee, ethics committee, or other institutional body to which data requests may be sent. If data are owned by a third party, please indicate how others may request data access. 6. Please include a separate caption for each figure in your manuscript. 7. Please include captions for your Supporting Information files at the end of your manuscript, and update any in-text citations to match accordingly. Please see our Supporting Information guidelines for more information: http://journals.plos.org/plosone/s/supporting-information. 

**Academic Editor Comments:**

Title

1. Please see my comment based on RCT-only study (result and miscellaneous).

Suggested title:

The efficacy and safety of acetazolamide in chronic mountain sickness：a systematic review and meta‐analysis of randomized placebo-controlled clinical trials

Abstract

1. It should include a background information, consider seeing other published systematic review articles at Plos One

2. Result section for insignificant result can be delivered concisely, not to specific as in the current version

3. It is better to add that more clinical trials with more participants are anticipated to further validated the findings (currently it is too

Introduction

1. erythrocytosis (EE) - should be excessive erythrocytosis (EE)

2. ....and an increase in hematocrit (Hct > 63%) is not universally adapted for everyone, it should be classified also as lower in women (approximately three times of Hb value) (https://www.frontiersin.org/journals/physiology/articles/10.3389/fphys.2020.00773/full)

3. Consider breaking and editing first paragraph of the introduction to two paragraphs, first one to describe CMS and second to inform about CMS cases

4. Third paragraph should describe available treatment options and describing acetazolamide role, also describing in more detail what is "there remains a lack of solid evidence-based support for its use" although it is already approved by the FDA

Methods

1. Add PRISMA long form and cite it

2. Add whether there is any age limitation or adult only inclusion in this study, also define what is "extended" (minimum period to be considered as a CMS)

3. Add any other types of human study, such as observational study (case-control, cohort, cross-sectional), case report, case series, etc

4. Cite which Endnote version that was used in this study

5. Cite Cochrane RoB tool utilized in this study, also Revman version

6. Add the reference stating I^2 50% as the threshold of heterogenous or homogenous data

Result

1. Since the authors stated that there is no blinding in reference 27, it cannot be considered as an RCT (just an experimental study, likely a quasi-experimental study), reconsider its inclusion

2. Divide its subsection based on similarity, such as laboratory parameters (pH, PaO2), etc

3. I agreed with reviewer 2 comments (why HCT is the primary outcome, not clinical score), please elaborate more or revise accordingly

4. Please do the GRADE analysis since the authors have strongly claimed that "we conclude that ACZ 250 mg is a safe, reliable, and low-cost treatment option for chronic mountain sickness. By reducing HCT, PaCO2, pH, and HCO3, and increasing PaO2, it improves respiratory and circulatory parameters in CMS patients and effectively treats CMS."

5. In accordance with a relatively high risk of bias in reference 27, it is recommended to remove it from the analysis (also because it is not RCT)

Discussion

1. Revise this: This study includes all "randomized controlled literature"

2. Please discuss more about the relevance of laboratory parameters and clinical status amelioration of CMS patients

3. Delete last sentence, it is redundant with the conclusion section (can be considered to be merged with it)

Conclusion

1. Same as abstract (3), basically to add further clinical trials or any other issues that should be encountered to strengthen the evidence

Table

1. Revise extraction table, it is distorted in the current version.

Miscellaneous

1. If the authors want to keep their statement in the title: The efficacy and safety of acetazolamide in chronic mountain sickness：a systematic review and meta‐analysis of randomized controlled trials, the authors should remove a study with reference 27 (it is not an RCT)

Citation

1. The authors should consider to make the references using referencing software, it is observed that there are several issues such as this: [26][27][28][29][30], which should be made as [26-30] AND writing of some journals such as J Appl pHysiol or Exp pHysiol, indicating inconsistent capitalization

Reviewers' comments:

Reviewer's Responses to Questions

**Comments to the Author**

1. Is the manuscript technically sound, and do the data support the conclusions?

Reviewer #1: Yes

Reviewer #2: Partly

Reviewer #3: Yes

2. Has the statistical analysis been performed appropriately and rigorously? 

Reviewer #1: Yes

Reviewer #2: Yes

Reviewer #3: Yes

3. Have the authors made all data underlying the findings in their manuscript fully available?

Reviewer #1: Yes

Reviewer #2: Yes

Reviewer #3: Yes

4. Is the manuscript presented in an intelligible fashion and written in standard English?

Reviewer #1: Yes

Reviewer #2: Yes

Reviewer #3: Yes

5. Review Comments to the Author

Reviewer #1: The efficacy and safety of acetazolamide in chronic mountain sickness：a systematic review and meta‐analysis of randomized controlled trials

A well performed and meticulous Meta-analysis of the effect of short term ACZ treatment of CMS.

Unfortunately, CMS is currently not clearly understood as it is assumed to be an isolated single disease present in some individuals at high altitude. Quite the contrary, CMS is in reality multiple diseases pulmonary, cardiac, renal, carotid body, hypoventilation and other alterations in a chronic hypoxia environment. Hence to treat all these multiple diseases as a sole individual pathology at high altitude gives rise to inadequate treatments. A poor diagnosis in medicine gives rise to ineffective and even deleterious effects of pharmaceutical strategies.

In order to adequately interpret CMS it is important to read “Redefining chronic mountain sickness: insights from high-altitude research and clinical experience”.

https://doi.org/10.1515/mr-2024-0036

In this Meta-analysis article based on the consensus statement CMS is “defined as hemoglobin concentration [Hb] ≥ 21 g/dL in men and ≥ 19 g/dL in women) and an increase in hematocrit (Hct > 63%), resulting in significantly increased blood viscosity, microcirculation disturbances, and even widespread organ damage and reduced blood flow velocity”

This is a misconception due to the poor understanding of CMS from the medical point of view as shown in the article above.

Approximately 81.6 million people worldwide live in high-altitude regions (>2500 meters) is not precise and should be reviewed as several papers point out to even over 100 million people.

Table 1 is disorganized (by software) and difficult to understand.

It is perfectly understandable that ACZ could not improve the clinical symptoms of CMS patients Fig. 4.

The results confirm ACZ effects but it questionable if it is a solution to CMS. Making pH acidic is not a good physiological response as cells require an optimal pH to function.

In the adverse events it becomes evident that the Richalet study was biased as compared to the other 4 studies, as no specific data is provided. Furthermore, upon literature review it is mentioned that in that study by Richalet there was actually a death episode, that was questionably attributed to other circumstances but this needs to be clearly stated.

Parestesia is indeed an important side effect of those using ACZ. It is very uncomfortable and often a sign of complain by those using it to prevent AMS.

The important analyses and evidence based observations regarding an increase of PV with the use of ACZ, gives rise to the question of confounding artificial benefits in the treatments.

with reference to “a new clinical CMS scoring system should be developed for interventional studies” it is evident that CMS scoring system is not efficient as it is not based in the true understanding of CMS.

Furthermore, the current treatment duration was relatively short, and it is highly questionable for long term use. Once the treatment is stopped after short use, the patients with CMS will return to their normal condition of increased Ht, lower PaO2, higher PaCO2 as this is a normal adaptive physiological response to disease in the chronic hypoxia environment.

In conclusion this excellent meta-analysis is actually a report of 5 different studies based on a wrong assumption of the short term treatment with ACZ of CMS interpreted as a sole entity with expected pharmacological results of the use of this inhibitor of carbonic anhydrase. In other words, it is excellent to show the effect of ACZ on the organism but questionable in its use as a permanent treatment of what is wrongly defined as CMS.

As the authors correctly mention in the limitations section, “more high-quality, large-sample, multi-center randomized studies are needed in the future to further validate the efficacy and safety of different doses of ACZ and long-term treatment for CMS”. It is highly probable that such a long term large population study will have unfavorable outcomes as ACZ is creating a pharmacological confounding response that satisfies the desperate attempt to reduce the Ht without a true understanding that the increase of Ht is actually a life saving physiological response to organ insufficiencies of multiple disease at high altitude. It may also have negative effects on longevity within those diseases at high altitudes.

Reviewer #2: The purpose of this systematic review was to evaluate the effectiveness and safety of acetazolamide(ACZ) in the treatment of chronic mountain sickness (CMS). The results of five randomized controlled trials were included, comprising a total of 137 subjects, with in the acetazolamide group and in the control group. The results show that ACZ 250 mg is a safe, reliable, and low-cost treatment option for chronic mountain sickness. It reduces hematocrit, arterial blood carbon dioxide pressure, pH, and bicarbonate concentration as well as increases arterial blood oxygen pressure as well as clinical scores of chronic mountain sickness.

The topic and the results are of interest, however, there are also some important concerns to be addressed.

1. Abstract/Methods, pH is the measure of H+ ion concentration, not the “potential of hydrogen”. The text should be revised accordingly.

2. Introduction, “oxygen receptors in the kidney” – I suppose authors mean hypoxia-induced factor (HIF). Strictly speaking, HIF should not be considered as the oxygen receptor because oxygen regulates HIF by modifying its hydroxylation through prolyl hydroxylase. Because O2 does not bind to HIF, the latter should not be named oxygen receptor.

3. Page 12, lines 1-4, the text suggests that ACZ has several mechanisms of action whereas all the effects listed here result from inhibiting carbonic anhydrase. The role of the enzyme in acid-base balance should be briefly decribed as well as the mechanisms through which its inhibition results in all effects listed in the sentence.

4. What are the specific aims of this meta-analysis? Meta-analyses are usually performed if there are discrepancies between the results of various experimental studies. No such discrepancies are mentioned in the Introduction and the rationale of performing the meta-analysis is questionable.

5. Exclusion criteria #1 – please specify which “other acute or chronic illnesses” were excluded and what approach was used for this purpose.

6. Why hematocrit rather than clinical score of CMS was the primary endpoint of interest?

7. Is there any relationship between duration of ACZ therapy and the outcomes?

Reviewer #3: I read with interest the manuscript entitled “The efficacy and safety of acetazolamide in chronic mountain sickness: a systematic review and meta‐analysis of randomized controlled trials”. The manuscript is well written and address an issue of interest with few quality data available.

Major comments

Inclusion criteria might include acetazolamide dose of 250mg/day

Length of treatment was not analyzed in depth. Inclusion criteria state that patients who have lived at high altitudes (≥2500 m) for an “extended period” were included. How long is an “extended period”

Length of treatment may also impact on some of the negative outcomes; therefore, it would be nice to have detailed median lengths of treatment by study.

The score (CMS) is composed of many variables, it would be possible that Acetazolamide may have beneficial effects only in some.

It would also be of importance to know if results can be sustained enough in time to revert cardiovascular adaptation. How long were the patients followed before assessing outcomes?

Minor comments

Document is not paginated

There are some mistakes in table 1 with headings in page 14 and body in the following page. The legend of table 1, is also missed in page 14.

6. PLOS authors have the option to publish the peer review history of their article (what does this mean? ). If published, this will include your full peer review and any attached files.

**Do you want your identity to be public for this peer review?** For information about this choice, including consent withdrawal, please see our Privacy Policy .

Reviewer #1: **Yes: ** Gustavo Zubieta-Calleja

Reviewer #2: No

Reviewer #3: No

---

## [Author Response · Author response to Decision Letter 1]

24 Dec 2024

Dear Editor:

Thank you for your letter and for the reviewers’ comments concerning our manuscript entitled “The efficacy and safety of acetazolamide in chronic mountain sickness：a systematic review and meta‐analysis of randomized controlled trials”.Those comments are all valuable and very helpful for revising and improving our paper,as well as the important guiding significance to our researches.We have studied comments carefully and have made correction which we hope meet with approval.The main corrections in paper and the responds to the editorial corrections and reviewer’s comments has been completed.Please refer to the document of “Response to Reviewers”for details.

Sincerely.

Feng Zhouzhou

---

## [Decision Letter · Decision Letter 1]

6 Feb 2025

The efficacy and safety of acetazolamide in chronic mountain sickness：a systematic review and meta‐analysis of randomized controlled trials

PONE-D-24-42748R1

Dear Dr. Feng,

We’re pleased to inform you that your manuscript has been judged scientifically suitable for publication and will be formally accepted for publication once it meets all outstanding technical requirements.

Kind regards,

Tungki Pratama Umar, M.D.

Academic Editor

PLOS ONE

Additional Editor Comments (optional):

Reviewers' comments:

Reviewer's Responses to Questions

**Comments to the Author**

1. If the authors have adequately addressed your comments raised in a previous round of review and you feel that this manuscript is now acceptable for publication, you may indicate that here to bypass the “Comments to the Author” section, enter your conflict of interest statement in the “Confidential to Editor” section, and submit your "Accept" recommendation.

Reviewer #2: All comments have been addressed

Reviewer #3: All comments have been addressed

2. Is the manuscript technically sound, and do the data support the conclusions?

Reviewer #2: Yes

Reviewer #3: Yes

3. Has the statistical analysis been performed appropriately and rigorously? 

Reviewer #2: Yes

Reviewer #3: Yes

4. Have the authors made all data underlying the findings in their manuscript fully available?

Reviewer #2: Yes

Reviewer #3: Yes

5. Is the manuscript presented in an intelligible fashion and written in standard English?

Reviewer #2: Yes

Reviewer #3: Yes

6. Review Comments to the Author

Reviewer #2: The manuscript has been revised according to the reviewers' comments. All concerns raised by the reviewers have been adequately addressed by the authors.

Reviewer #3: (No Response)

7. PLOS authors have the option to publish the peer review history of their article (what does this mean? ). If published, this will include your full peer review and any attached files.

**Do you want your identity to be public for this peer review?** For information about this choice, including consent withdrawal, please see our Privacy Policy .

Reviewer #2: No

Reviewer #3: No

---

## [Editor Report · Acceptance letter]

PONE-D-24-42748R1

PLOS ONE

Dear Dr. Feng,

I'm pleased to inform you that your manuscript has been deemed suitable for publication in PLOS ONE. Congratulations! Your manuscript is now being handed over to our production team.

Kind regards,

on behalf of

Dr. Tungki Pratama Umar

Academic Editor

PLOS ONE